# Molecular, Histological and Histochemical Responses of Banana Cultivars Challenged with *Fusarium oxysporum* f. sp. *cubense* with Different Levels of Virulence

**DOI:** 10.3390/plants11182339

**Published:** 2022-09-07

**Authors:** Anelita de Jesus Rocha, Julianna Matos da Silva Soares, Fernanda dos Santos Nascimento, Adailson dos Santos Rocha, Vanusia Batista Oliveira de Amorim, Andresa Priscila de Souza Ramos, Cláudia Fortes Ferreira, Fernando Haddad, Edson Perito Amorim

**Affiliations:** 1Departamento de Ciências Biológicas, Universidade Estadual de Feira de Santana, Feira de Santana 44036-900, Bahia, Brazil; 2Departamento de Ciências Biológicas, Universidade Federal do Recôncavo da Bahia, Cruz das Almas 44380-000, Bahia, Brazil; 3Embrapa Mandioca e Fruticultura, Cruz das Almas 44380-000, Bahia, Brazil

**Keywords:** fusarium wilt, *Musa* spp., gene expression, plant resistance

## Abstract

Fusarium wilt caused by *Fusarium oxysporum* f. sp. *cubense* (Foc) is the most limiting factor in the banana agribusiness worldwide. Therefore, studies regarding pathogen attack mechanisms, and especially host defense responses, in this pathosystem are of utmost importance for genetic breeding programs in the development of Foc-resistant banana cultivars. In this study, analysis at the molecular, histological and histochemical levels of the *Musa* spp. x Foc interaction was performed. Three Foc isolates representative of race 1 (R1), subtropical race 4 (ST4) and isolate 229A, which is a putative ST4, were inoculated in two Prata-type cultivars (Prata-Anã and BRS Platina) and one cultivar of the Cavendish type (Grand Naine). Of seven genes related to plant–pathogen interactions, five were overexpressed in ‘BRS Platina’ 12 h after inoculation (HAI) with Foc R1 and ST4 but had reduced or negative expression after inoculation with Foc 229A, according to RT–qPCR analyses. While hyphae, mycelia and spores of the Foc 229A isolate grow towards the central cylinder of the Grand Naine and Prata-Anã cultivars, culminating in the occlusion of the xylem vessels, the BRS Platina cultivar responds with increased presence of cellulose, phenolic compounds and calcium oxalate crystals, reducing colonization within 30 days after inoculation (DAI). In general, these data indicate that the cultivar BRS Platina has potential for use in banana-breeding programs focused on resistance to Foc tropical race 4 (TR4) and in aggregating information on the virulence relationships of the Foc pathogen and the defense responses of banana plants after infection.

## 1. Introduction

Fusarium wilt, caused by *Fusarium oxysporum* f. sp. *cubense* (EF Smith) Snyder e Hansen (Foc), is an imminent threat to banana and plantain production [1,2,3]. Although it originated in Southwestern Asia, where it coevolved with bananas, the pathogen was identified for the first time in Australia in 1876, from where it spread, causing the destruction of more than 40,000 ha of the cultivar Gros Michel [4,5,6]. At the time, the cultivar Gros Michel was replaced by Cavendish cultivars, resistant to race 1 of the pathogen, which began to substantially make up the export trade of the fruit [7,8]. However, around 1990, Cavendish plantations began to be heavily affected by tropical race 4 (TR4), a new highly virulent strain that devastated Southeast Asian plantations and has spread to several regions of the world [7].

Currently, tropical race 4 is responsible for the destruction of banana plantations and creating unsuitable soil conditions on almost all continents, and is present in Asia, Africa, Indonesia and South America [9,10,11,12]. Conversely, subtropical race 4 (ST4) is characterized by causing symptoms in Cavendish cultivars in subtropical regions when there are environmental stress factors, such as temperature extremes or water deficits [3,7,8,13].

In infested fields, there are still a lack of economically viable measures for disease management. This is because Foc is a soil-borne pathogen that produces chlamydospores, resistant spores that remain viable in the soil for many years; a saprophytic habit allows this pathogen to survive in dead organic matter in the absence of a host [9,10,14,15]. The use of chemical treatments raises environmental and health concerns, and although biological control is expanding, its efficacy has not yet been demonstrated under field conditions [16,17]. Therefore, genetic improvement and selection for resistance is the best option for a effective and sustainable management. Many breeding programs of *Musa* sp. in different research centers around the world are focused on obtaining resistant cultivars from crosses. 

Thus, information obtained after sequencing the genome of wild diploid banana ‘Pahang’ is in use, aiming to achieve more data from the transcriptome of Foc-infected bananas and to better understand the molecular mechanism of Fusarium wilt resistance applied in banana genetic breeding [6,18]. Knowledge of plant–pathogen interaction genetics, focused on understanding both pathogen attack mechanisms and plant defense responses, has been explored in many studies, mainly from transcriptome data of bananas infected with Foc and differentially expressed genes (DEGs) [12,19,20,21,22]. Thus, many studies seek to differentiate the defense responses of banana plants to infection by different Foc isolates, mainly to understand the susceptibility of the Cavendish cultivars to Foc TR4 [23]. 

It is well-known that pathogens can secrete effectors to regulate the immune response of plants, while plants can also produce specific receptor-like protein kinases (RLKs) to recognize and fight pathogen infection [24,25,26]. Many RLK genes have been associated with responses of banana plants to infection to Foc TR4, and other genes, such as chitin receptor elicitor kinase 1 (CERK1), flagellin-sensitive 2 (FLS2), serine/threonine-protein kinase (PBS1), transcription factor WRKY 22 (WRKY22), pathogenesis-related proteins (PR-1), chitinase, lipoxygenases (LOX), jasmonate (ZIM), domain protein (JAZ), glutathione-S-transferase (GST) and cellulose synthase, may contribute to differentiate the virulence between the Foc R1 and TR4 races and account for the difference in the host resistance response [21,23,27]. 

Although there are a substantial amount of data on banana defense responses in the interaction with the Foc pathogen, knowledge of the pathogenic molecular mechanism and its interaction with the host has not yet been fully elucidated. In addition, there is no cultivar available that is immune to Foc TR4, or with a desirable baseline resistance level that can replace Cavendish cultivars. Therefore, all efforts to understand the mechanisms of attack of the pathogen, such as the defense responses, can bring valuable contributions to banana genetic breeding, mainly by identifying global patterns of gene expression, influenced by infection of different races. Therefore, the present study aims to evaluate the infectious process of Foc isolates with different virulence profiles in the roots of banana cultivars at molecular level by quantifying the expression of resistance genes by qRT-PCR over time and at histological and histochemical levels. These data will bring new contributions to the development of fusarium wilt-resistant banana cultivars and expand knowledge about the mechanisms involved in plant–pathogen interactions.

## 2. Results

### 2.1. Expression Profile of Defense Genes in Response to Different Foc Isolates

To verify the interaction of banana cultivars BRS Platina, Grand Naine and Prata-Anã with Foc R1, ST4 and 229A isolates at the molecular level, a total of seven genes potentially involved in the plant–pathogen interaction (ATL, CESA7, CHI, LOX, PI206, WRKY22, PR1) related to disease resistance were selected, and qRT-PCR validation was performed. The set of genes analyzed showed different expression patterns for cultivars infected with different isolates. In general, the cultivar BRS Platina demonstrated an upregulated expression level for most genes in the first 12 h after inoculation with Foc R1 and ST4. The results of the expression pattern of these genes were consistent and showed correlation with fluorescence microscopy analyses for the detection of cellulose, with scanning electron microscopy analyses and with the symptomatic analyses, considering that for both analyses the cultivar BRS Platina demonstrated greater defense responses against the different Foc isolates.

The Auxin transporter-like protein 1 (ATL) gene, related to auxin response, was upregulated in the cultivar BRS Platina at all evaluation times, starting with the highest levels at 12 HAI, mainly for Foc R1 and ST4 isolates, but for Foc 229A, it was downregulated, mainly at 48 HAI (Figure 1). Conversely, in relation to the cultivars Prata-Anã and Grand Naine, the ATL gene was gradually upregulated over time, with the highest levels of expression at 24 or 48 HAI (Figure 1). The upregulated expression of ATL in the resistant cultivar at 12 HAI mainly for Foc R1 and ST4 indicates a possible involvement in rapid defense responses after infection. The Cellulose synthase gene A catalytic subunit 7 (CESA7), which is part of different classes of proteins related to the synthase cellulose complex, showed an oscillation in the level of expression according to inoculation times, and for the cultivar BRS Platina, inoculated with Foc R1, an upregulated expression was observed only at 12 HAI. In relation to Foc ST4, there was an upregulated expression at all evaluation times; for isolate Foc 229A, the expression was upregulated in 12 and 24 HAI (Figure 1). In the cultivar Prata-Anã inoculated with Foc R1, the CESA7 gene was only upregulated at 48 HAI, in comparison to ST4, whose gene expression was upregulated in 12 HAI, downregulated in 24 HAI and again upregulated in 48 HAI. For the cultivar Grand Naine, the CESA7 gene was upregulated at inoculation with Foc R1, with the highest levels of expression being downregulated at 12 HAI. As with Foc ST4 and 229A, gene expression was upregulated at 48 HAI (Figure 1).

The Chitinase gene (CHI) that encodes enzymes that catalyze chitin hydrolysis offering antifungal activity was upregulated mainly in the cultivar BRS Platina inoculated with Foc R1 in 12 HAI; and with inoculation with Foc ST4 it was upregulated in 12 and 48 HAI; but for interaction with Foc 229A, the expression level was lower in comparison to the cultivar Prata-Anã (Figure 1). In the cultivar Grand Naine, the CHI gene was downregulated in all hours after inoculation with Foc R1, and in inoculation with Foc ST4 and 229A it had an upregulated expression at 12 and 48 HAI. For the cultivar Prata-Anã inoculated with Foc R1, the expression of the CHI gene in all hours after inoculation was downregulated; and for inoculation with Foc ST4, there was an upregulated expression at 12 and 48 HAI; while in relation to inoculation with Foc 229A, the CHI gene presented an upregulated expression superior to all cultivars in 12 HAI, but this expression was downregulated at 24 and 48 HAI (Figure 1). Thus, the CHI gene seems to be associated with a rapid defense response in the cultivar BRS Platina for Foc R1 and ST4, but not for resistance to Foc 229A.

The Lipoxygenase gene (LOX) was upregulated in the cultivar BRS Platina inoculated with Foc R1 and ST4 isolates in 12 HAI, reducing expression over time, unlike the cultivars Grand Naine and Prata-Anã, where the expression was downregulated in all hours after inoculation of these isolates (Figure 2). Regarding inoculation with Foc 229A, the LOX gene was upregulated in the first 12 HAI mainly for cultivar Prata-Anã and in 48 HAI for the cultivars Prata-Anã and Grand Naine (Figure 2). Similarly, to the expression profile of the ALT and CHI genes, the LOX gene seems to play essential functions. In the initial defense responses of cultivar BRS Platina to Foc R1 and ST4, but it does not seem to be related to the resistance response of the cultivar Grand Naine to Foc R1, since its expression was downregulated. The Putative Disease resistance response protein 206 gene (PI206) was upregulated exclusively in the cultivar BRS Platina inoculated with Foc R1 and ST4 especially at 12 HAI, considering that the expression of this gene was downregulated in the cultivars Grand Naine and Prata-Anã for all hours after inoculation of the isolates (Figure 2). At inoculation with Foc 229A, there was no expression of the PI206 gene considered upregulated for any of the cultivars (Figure 2).

Transcription factor WRKY (WRKY22) was upregulated in the resistant cultivar BRS Platina inoculated with Foc R1 and ST4 at 12 HAI, and over time, the expression was reduced. In the interaction with Foc 229A, it was upregulated only at 12 HAI, and the expression was downregulated at 24 and 48 HAI (Figure 3). In the cultivar Grand Naine inoculated with Foc R1, the expression of the WRKY22 gene was upregulated at 12 HAI and downregulated at 24 and 48 HAI; already in relation to inoculation with Foc ST4, the expression was upregulated at 12 and 48 HAI, and downregulated at 24 HAI; and in relation to inoculation with Foc 229A, the expression was upregulated, especially at 12 and 48 HAI (Figure 3). In the susceptible cultivar Prata-Anã inoculated with Foc R1, the WRKY22 gene was upregulated in all hours after inoculation, but the expression was lower in comparison to the cultivar BRS Platina. At inoculation with Foc ST4 and 229A, the expression was upregulated at 12 and 48 HAI and downregulated at 24 HAI (Figure 3). The Pathogenesis-related proteins 1 (PR1) gene encodes pathogenesis-related proteins and, in general, had the highest levels of upregulated expression in relation to the other genes analyzed for all cultivars (Figure 3). At 12 HAI, all cultivars showed upregulation of the PR1 gene when inoculated with Foc R1 and Foc ST4 and 229A, but the expression levels in the cultivar BRS Platina were higher when inoculated with Foc ST4 and lower in interactions with Foc 229A (Figure 3). For interactions with the isolate ST4 at 12 HAI, the cultivar Prata-Anã presented the highest levels of overexpression, while in the cultivar Grand Naine, the levels of relative expression were higher only at 48 HAI (Figure 3).

### 2.2. Histological and Histochemical Responses

The production of cellulose was verified by staining samples from all treatments at 12 HAI with Calcofluor White dye where positive samples emitted secondary blue-white fluorescence in the parenchyma and central cylinder. For the BRS Platina cultivar, the emission of fluorescence was reflected in the inoculation with isolates Foc R1, ST4 and 229A (Figure 4). These observations were consistent with the CESA7 gene expression data, related to cellulose synthesis, since the cultivar BRS Platina had higher expression of this gene in 12 HAI (Figure 1), as well as with the symptomatological analyses, considering the lower level of symptoms of the disease in the cultivar BRS Platina inoculated with the isolates compared to the cultivars Grand Naine and Prata-Anã. Therefore, cellulose is an apparently efficient defense response for the cultivar BRS Platina. However, in the cultivars Grand Naine and Prata-Anã, the fluorescence was effused only after inoculation with Foc ST4. In general, the control samples did not present high fluorescence levels (Figure 4).

The ferric chloride test to detect phenolic compounds produced small dots in the tissue with a dark brown color in the rhizomes at 12 HAI (Figure 5). All the cultivars showed positive results for the presence of phenolic compounds in samples inoculated with Foc R1, ST4 and 229A, and a strong indication of phenolic compounds in the control samples was observed, which is due to environmental changes or a constitutive basal production of this compound in each cultivar (Figure 5). The presence of phenolic compounds in the cultivar BRS Platina was more significant when inoculated with Isolate ST4 (Figure 5). In the cultivar Grand Naine, the presence of phenolic compounds was higher in the interaction with isolate 229A and in the cultivar Prata-Anã when inoculated with Foc R1 and Foc 229A isolates (Figure 5). Considering that the presence of fusarium wilt symptoms is associated with higher oxidation of phenolic compounds, these observations are consistent with the symptomatological analyses performed at 90 DAI, considering that the cultivar BRS Platina demonstrated a higher level of symptoms for the Foc ST4 isolate, cultivar Grand Naine for Foc 229A and Prata-Anã cultivar for both isolates.

With the clarification of the tissues and staining of Foc structures inside the roots at 90 days after inoculation, it was possible to determine that all cultivars were positive for the presence of pathogen structures such as hyphae and spores, indicating that penetration occurs inside the roots both in resistant cultivars and susceptible cultivars. Thus, in the resistant cultivar BRS Platina, the presence of hyphae inside the tissue in the interaction with Foc R1 and 229A, and spores in the interaction with Foc ST4 (Figure 6), were observed. There were pathogen spores in the cultivar Grand Naine inside the tissues even for the interaction between the cultivar Grand Naine and Foc R1, for which there were no symptoms characterizing an immunity response. In this cultivar, the presence of hyphae and spores in the interaction with Foc ST4 and 229A was also detected (Figure 6). For the cultivar Prata-Anã, hyphae were found in the tissue and there were spores in the interaction with Foc R1, ST4 and 229A (Figure 6).

### 2.3. Analysis of the Interaction by Scanning Electron Microscopy

An analysis using scanning electron microscopy indicated that at 48 HAI the cultivar BRS Platina responded to infection by Foc ST4 and 229A with intense production of calcium oxalate crystals (Figure 7). This response was not observed in the interaction with Foc R1, but the tissues were intact and without the presence of fungal growth, indicating probable delay in the penetration of this isolate in comparison to the others. No structural defense responses or pathogen growth were observed in the Grand Naine cultivar inoculated with this pathogen at 24 HAI (Figure 7). In the Prata-Anã cultivar, growth of fungal mycelium and early obstruction of the conducting vessels were observed in the interactions with Foc R1 and ST4, and in the interactions with Foc 229A, pathogen sporulation was observed, confirming the greater virulence of this isolate by its ability to rapidly grow and reproduce within the tissue in the susceptible cultivar. Structural changes or traces of the pathogen in the control samples of the three cultivars were not observed (Figure 7). 

The extent of pathogen infection in different cultivars was analyzed via SEM. At 30 DAI, fungal hyphae extensively colonized the epidermis, collenchyma, cortical and medullary parenchyma and vascular bundle of the Prata-Anã susceptible cultivar inoculated with isolates R1, ST4 and 229A, with obstruction of the xylem vessels caused by the presence of large amounts of mycelium and spores linked to late defense responses mainly in the interaction with Foc R1 (Figure 8). These data confirm the susceptible phenotype of the Prata-Anã cultivar observed in the symptom evaluations at 90 DAI as they may be associated with the downregulated expression of some important resistance genes in plant–pathogen interaction such as the CHI (Figure 1), LOX and PI206 genes (Figure 2).

In contrast, in the BRS Platina cultivar, the occlusion of conducting vessels was observed only when inoculated with the ST4 isolate (Figure 8), but the pathogen did not reach the central cylinder or xylem vessels; it only reached the parenchyma and collenchyma tissues that surround the central cylinder, demonstrating that there is no growth of the pathogen in a timely manner to reach the rhizome. In this cultivar, occlusion or growth structures were not observed in plants inoculated with Foc R1 and 229A at 30 DAI (Figure 8). This result is consistent with the upregulated expression of important defense genes in this cultivar such as CESA7 and CHI (Figure 1) and LOX and PI206 (Figure 2), which may have culminated in the lowest rates of symptoms in relation to isolates at 90 DAI (Figure 9). The Grand Naine cultivar presented tissue occlusion in the parenchyma and collenchyma that extended to the central cylinder, with the presence of fungal mycelium at 30 DAI when inoculated with the ST4 and 229A isolates. The root tissues of this cultivar had no trace of the pathogens in the interactions with the R1 isolate (Figure 8), and this may be related to the resistant profile of race 1 of the pathogen, although at 90 DAI it is possible to identify the presence of hyphae by the method of clarification and staining of fungal structures, which indicates a postpenetration defense response in relation to Foc R1 (Figure 6). 

### 2.4. Symptoms 

In addition to molecular, histological and histochemical analyses and to verify the efficiency of the inoculation method, 10 plants of each cultivar for each isolate were evaluated as to the presence of symptoms in cross-sections of the rhizome at 90 DAI. As expected for a susceptibility profile, the cultivar Prata-Anã presented characteristic symptoms of wilting with the death of some plants, mainly when interacting with the isolate ST4. Symptoms were also evident in the cross-section of the plant rhizome, where a reddish-brown color was observed with infection points starting from the extremities and reaching the central cylinder until its complete obstruction (Figure 9). It was possible to identify and quantify the aggressiveness of the isolates by the index of internal symptoms that are demonstrated by the heat map in Figure 9, where the colors of intense red indicate higher indices, and those of intense green indicate lower indices. For the Prata-Anã cultivar, the indices were 90%, 98% and 59% for isolates of Foc R1, ST4 and 229A, respectively (Figure 9). As Cavendish cultivars are resistant to Foc R1 in the cultivar Grand Naine, the Foc R1 isolate did not cause the disease; the values of the symptom index for the ST4 isolate were 14%, while those for isolate 229A were 70% (Figure 9). 

In the cultivar BRS Platina, there was no discoloration of the rhizome associated with practically any of the isolates. The index of internal symptoms in ‘BRS Platina’ inoculated with isolate ST4, 229A and Foc R1 was 27%, 9% and 11%, respectively (Figure 9). These data therefore show that in fact the upregulated genes in the cultivar BRS Platina, as well as the cellulose related to the cell wall seen by fluorescence microscopy may play important roles in the delay of infection by the pathogen seen by SEM analyses, which culminated in this phenotype of resistance to Foc isolates with a lower symptom index of the cultivars Grand Naine and Prata-anã (Figure 9). Overall, heatmap-based cluster analysis showed that the cultivars Grand Naine and Prata-Anã grouped in a single cluster, different from the cultivar BRS Platina, which was much more distant with a disease index of less than 30% considering all isolates (Figure 9). This implies that the Foc 229A isolate was more aggressive in the cultivar Grand Naine than in the cultivars Prata-Anã and BRS Platina, which confirms its virulence profile.

## 3. Discussion

Seven genes related to plant–pathogen interaction pathways were validated by real-time quantitative polymerase chain reaction (qRT–PCR) from root samples of BRS Platina, Grand Naine and Prata-Anã cultivars following interactions with Foc R1, ST4 and 229A isolates. Differences were observed in the level of gene expression in all the cultivars according to the isolate, where high levels of overexpression were generally associated with interactions with the ST4 isolate, and lower levels were associated with interactions with Foc 229A. These results confirm that differences in the virulence of the isolates may contribute to changes in the host resistance response and that the expression of defense-related genes may alter the efficacy of pathogen virulence mechanisms in some cultivars [29].

Plant hormones play a key role in the interaction between plant development and the environment in order to create a signal pathway that helps to mold its architecture and at the same time prepare it to respond to certain stresses appropriately. Auxins or their signal pathways modulate resistance in plants to diseases directly and indirectly. The direct effects of auxins are interference in the circuit of signalization, and indirectly they may alter the progress of the disease during the plant–pathogen interaction due to their effect on plant development [30,31]. In the present study, the ATL gene, which is related to auxin production and signaling, seems to play an important role in interactions of the BRS Platina cultivar with Foc R1 and ST4, considering that its expression was increased at all times, except in the interaction with Foc 229A, for which a reduction was observed. These results are in agreement with the evaluation by Costa [32], who found that this gene was exclusively associated with the ‘BRS Platina’ genome when inoculated with the isolate Foc R1. In addition, auxins are known to be important phytohormones for plant growth and disease resistance. Some studies have suggested that pathogens could increase auxin biosynthesis in a plant to alter the growth and development of the plant in its favor [31]. However, our symptomatology results indicate that the increased expression of the ATL gene did not contribute to greater susceptibility of the BRS Platina cultivar to the ST4 isolate, nor did its reduction seem to be associated with resistance in the other cultivars, considering the isolate.

Besides offering structural support and a passive barrier against pathogens, the cell wall controls cell expansion and is involved in the exchange of water and substances during plant development. It is also considered a reservoir of antimicrobial components, and a source of signaling molecules where alterations in the cell wall influence growth and the network of responses to stress, especially during the response to pathogens that survive in the apoplast, as is the case with *Fusarium oxysporum* in bananas [33]. In the case of chitin β-(1–4)-poly-N-acetyl-D-glucosamine), it is also a structural compound of the exoskeleton of crustaceans and is abundantly distributed in the cell walls of fungi [34]. Therefore, chitinase (CHI gene), an enzyme capable of passing through the cell walls of fungi, may offer plants good protection against pathogens. The CESA7 gene, which is part of a family of genes required for cellulose biosynthesis in the secondary cell wall, and the CHI gene, which regulates chitinase production, were upregulated in the roots of the banana cultivar BRS Platina at 12 HAI with Foc R1 and ST4 and downregulated in the Prata-Anã and Grand Naine cultivars at almost all times after inoculation with the isolates (Figure 2). After infection by pathogens, it was suggested that the plant responds with the synthesis of new carbohydrates, especially callose and cellulose, which are added to the interior of the cell wall adjacent to the infection, in order to contain the advance of the fungus [26,32]. 

In this study, both the expression of the CESA7 gene and the presence of cellulose detected by fluorescence microscopy were reduced in the interaction with the 229A isolate, which at the end of the experiment was considered more aggressive in the Prata-Anã and Grand Naine cultivars (Figure 2). Given the aggressiveness of this isolate, this finding may be associated with the fact that fungal and bacterial pathogens can produce cellulases as virulence factors, which have the ability to break down cellulose in the host cell wall [33,35]. As for the cultivars Prata-Anã and Grand Naine, the presence of cellulose by fluorescent microscopy was not consistent with the data of expression of gene CESA7, being present only after inoculation with Foc ST4 (Figure 4), whereas the expression of gene CESA7 was upregulated in cultivar Prata-Ana inoculated with Foc 229A, and was not differential in cultivar Grand Naine inoculated with Foc ST4 (Figure 1). The CESA genes for the respective CESA complexes are highly co-regulated since many CESA genes are reported as playing key roles in cellulose synthesis in the secondary cell wall, such as the CESA8 and CESA6 genes [36]. Although these genes were not evaluated in this study, a hypothesis is that there might have been some changes in the combined regulation of these genes in cultivars Prata-Anã and Grand Naine in the interaction with isolates Foc R1 and 229A, culminating in the absence of fluorescence in Figure 4. 

Chitinases target components of the fungal cell wall, breaking down β-1,3-glucans and chitin, which are important pathogen virulence factors [29,37]. The CHI gene was more highly expressed in the BRS Platina cultivar than in the Prata-Anã and Grand Naine cultivars at 12 HAI, which may be closely related to the resistance of this cultivar in the early stage of infection [32]. In contrast, with the 229A isolate, there was a reduction in the expression of the CHI gene in all the cultivars, which may be related to the greater virulence of this isolate. Ding et al. [37] demonstrated that the null mutation of three mitogen-activated protein kinase (MAP) genes led to a substantial attenuation of fungal virulence in bananas, especially in the regulation of genes encoding the production of chitin, peroxidase, beauvericin and fusaric acid, demonstrating that these genes can actually alter the response mechanisms of infected plants. In another study, Li et al. [29] demonstrated that the CHI gene was more induced in the mutant resistant to Foc TR4 ‘Nongke No. 1’ (NK) than in the susceptible cultivar Baxi (BX) at 27 HAI, which may be closely related to the resistance of NK in the initial phases of infection. Furthermore, that study showed that fungal genes that express chitin synthase (CHS), necessary for *Fusarium oxysporum* pathogenesis, had higher expression in BX than in NK and may indicate higher pathogenicity in a susceptible infected cultivar than in a resistant one. 

The lipoxygenase (LOXs) are enzymes of natural occurrence amply distributed in plants and animals. These enzymes are capable of deoxygenizing unsaturated fatty acids, which leads to lipoperoxidation of biological membranes. This process causes the synthesis of signaling molecules and also may alter cell metabolism. LOXs are known for being involved in the apoptosis (programmed cell death) pathway and promote the biosynthesis of jasmonic acid (JA), acting as a biomarker of stress against fungi, bacteria, pests and abiotic stresses [38,39,40,41]. In this study, there was upregulation of the LOX (lipoxygenase) gene in the BRS Platina cultivar at all hours after inoculation with isolates R1 and ST4, but for isolate 229A, the regulation was reduced by 12 HAI. In the Prata-Anã and Grand Naine cultivars, this expression was downregulated in all evaluations, except at 12 HAI and 48 HAI (Figure 3). A RT–qPCR analysis performed by Liu et al. [42] revealed that several genes of the LOX family may increase banana resistance to Foc TR4 by regulating the jasmonic acid (JA) pathway. In another study, Li et al. [43] found that the high expression of LOX was related to greater resistance to Foc TR4 in a mutant, and Li et al. [44] also found that LOX1.1-3 and LOX2.3 were significantly induced in a resistant strain of *Musa yunnanensis* during early infection with Foc TR4.

The PI206 gene was overexpressed only in the BRS Platina cultivar inoculated with Foc R1 and ST4 at 12 HAI, but there was no upregulation in any of the cultivars inoculated with the 229A isolate (Figure 3). This gene is part of the leucine-rich repeat domain of the nucleotide binding site (NBS-LRR) and was identified exclusively in the ‘BRS Platina’ genome via RNAseq analysis [32]. NB-LRR genes are the largest group of plant R genes and play important roles in the perception of extracellular immunogenic patterns that trigger defense responses to prevent the spread of a pathogen [45,46]. An analysis of the entire genome of the LRR-RLP gene family in wild banana *Musa acuminata* ssp. *malaccensis* identified several candidate genes for Fusarium wilt resistance [46]. In other studies, NBS-LRR genes were strongly induced after inoculation of TR4 in resistant cultivars, indicating that these genes play important roles in banana against Foc infection [20,43,47].

Transcription factors (TFs) are part of the machinery of the defense response to stress through regulation of a complex system of genes in plants. The WRKY superfamily of TFs is the seventh largest in plants with flowers and is a promising candidate for plant breeding due to rigid regulations involving the specific recognition and link of WRKYs to forward promoters. WRKYs orchestrate molecules in plants and provide multiple simultaneous responses where activation or repression occurs through the recognition of W-box sequences present in promoter sequences of target genes [48,49]. Many WRKY functions involved in defense responses against biotic and abiotic stress have been studied [50,51,52,53]. 

In the present study, the WRKY22 gene, belonging to a family of transcription factors known to play an important role in resistance to biotic stress, had a higher relative expression profile at all hours after infection in the BRS Platina cultivar inoculated with Foc R1, ST4 and 229A (Figure 4). It is known that transcription factors are part of many signaling pathways that regulate plant defense responses. The superfamily of WRKY transcriptional regulators in addition to regulating the expression of defense genes also regulate response pathways to diseases regulated by salicylate and jasmonate [54,55]. In one study, seven WRKY genes, including one WRKY22 gene, were involved in the plant–pathogen pathways and were twice as high in the Pahang cultivar as in the Cavendish cultivar, suggesting that the expression of these WRKY genes may be associated with constitutive defense mechanisms [22]. WRKY56 and WRKY75 genes were also associated with the response of the resistant Cavendish banana mutant ‘Nongke No 1’ to Foc TR4 [43].

Pathogen-related proteins are-well studied in the literature, and PR1, among the 17 families, is the dominant group induced by pathogens or salicylic acid used as a marker of pathogen-induced systemic acquired resistance (SAR) [56,57]. Its role as an indispensable component of native immune responses in plants under biotic or abiotic stress and its interaction with the inhibition of pathogen effectors are also reported [58,59]. In our study, the PR1 gene presented an overregulated expression profile compared to all the other genes evaluated for all cultivars inoculated with Foc R1 and ST4, except for the 229A isolate, whose expression of the PR1 gene was reduced, especially in the cultivar BRS Platina (Figure 4). High levels of expression of the PR1 gene in response to Foc TR4 are reported in the literature [22,60,61].

In our study, the BRS Platina cultivar responded to the Foc R1, ST4 and 229A isolates with a greater presence of cellulose in its tissues than that observed for the Prata-Anã and Grand Naine cultivars. This result is consistent with the expression data of the CESA7 gene, which is related to cellulose synthesis. These data suggest that cellulose is not an important component in the defense immune response of the Grand Naine cultivar to the Foc R1 isolate. Phenolic compounds seem to be extensively produced in the roots of the Prata-Anã cultivar at 48 HAI. It has been suggested that the accumulation of phenolic compounds indicates the sensitivity of the plants to the pathogen and the attempt to protect themselves by the formation of structural barriers [62]. Therefore, this result may be related to the extreme susceptibility of this cultivar during interactions with the studied isolates. On the other hand, phenolic assays indicated that tolerance to Foc ST4 may be linked to the increase in phenolic compounds associated with the cell wall [63]. It is believed that this may also be one of the resources used by the resistant cultivar BRS Platina as a defense response in relation to the ST4 isolate, since there was an increase in this compound.

SEM data showed that calcium oxalate crystals are produced in abundance in the BRS Platina cultivar inoculated with Foc ST4 and 229A and can play important roles in resistance. In contrast, this type of response was not observed in the interactions between any of the isolates with the cultivars Prata-Anã and Grand Naine, and even only at 48 HAI, pathogen structures such as hyphae, spores and mycelium were found inside the roots of the cultivar Prata-Anã (Figure 7). Although there are few studies reporting the presence of calcium oxalate crystals as a defense response in interactions between *Musa* spp. x Foc, it is believed that these crystals play an important role, especially because the degradation of single crystals can produce reactive oxygen species, which have been extensively related to the response to infection by pathogens and have also been related to the inhibition of infection [32,64]. In addition, in the staining data of fungal structures, after root clarification, all the isolates penetrated the tissues of susceptible and resistant cultivars, which is in agreement with the evaluations performed by Dong et al. [27], who observed that both Foc R1 and Foc TR4 could penetrate the root epidermis and invade the xylem vessels of Cavendish cultivars.

At 30 DAI, SEM analysis showed that the cultivar Prata-Anã behaves as a plant with a high level of susceptibility to isolates R1, ST4 and 229A; that is, progression of the invasion by the pathogen and clear failures in the responses of defense culminated in the collapse of the xylem vessels, which was observed at 90 DAI. In contrast, in the interactions of the BRS Platina cultivar with the ST4 isolate, the vessel occlusion in the xylem was limited to the parenchyma and had not yet reached the central root cylinder at 30 DAI, which suggests that the containment of the pathogen advanced over time. Regarding the Grand Naine cultivar, the interaction with isolate 229A resulted in the occlusion of vessels and mycelium, indicating that the pathogen continued to expand, advancing towards the central cylinder, which explains why this isolate contributed 70% of the internal symptom index at 90 DAI for this cultivar. In a study that followed the colonization pathway of Foc ST4 in susceptible banana genotypes, Warman and Aitken [65] also found a 30 dpi Cavendish root sample, with macroconidia forming outside the root surface 

All this information considered together allows us to infer that different defense response mechanisms are taken by the BRS Platina, Grand Naine and Prata-Anã cultivars based on the virulence of the isolates. Thus, it is reported that the genes CESA7, ATL, PI206, WRKY22, PR1, CHI and LOX may not play important functions in the first hours of interactions in the immune defense response of the Grand Naine cultivar to the Foc R1 isolate. Other mechanisms may confer race-specific resistance to this cultivar of the Cavendish subgroup, given that the complexity of the defense system of these cultivars has been documented in other studies [6,43]. According to Dong et al. [23], after infection by Foc R1, the synthesis pathways of lignin and flavonoids are enriched in Cavendish cultivars, and when measuring the expression patterns of defense-associated genes, five overexpressed genes were found that can cause hypersensitive cell death after infection.

On the other hand, it was demonstrated that the seven genes studied may play essential functions in the defense response of the BRS Platina cultivar in the interaction with the three Foc isolates, especially in the first hours after inoculation, and these data were supported by SEM and histochemical analyses and symptomatological data. In a previous study, it was determined that the type of resistance response involved in the interaction between the Foc 0801 isolate (race 1) and the BRS Platina cultivar is based on cell wall modifications, such as the formation of a healing zone, the presence of tylose and oxalate crystals, and lignification [32,66]. Therefore, this shows that this resistance is not the same resistance of Cavendish cultivars to Foc R1 isolates but a quantitative type of resistance that can be very well-explored, especially in a culture system based on integrated management. In addition, all the genes included in this study have been reported to be important in the response of banana genotypes to Foc TR4 infection. 

When interacting with isolate 229A, which is considered more aggressive than the other isolates, the cultivar BRS Platina and the other cultivars had reduced levels of gene expression, especially for PI206 and PR1. Isolate 229A was initially isolated from the Prata-Anã cultivar in 2014 in farms in the municipality of Miracatu in the state of São Paulo in Brazil [67]. In a recent study that characterized Foc populations of several regions of Brazil [68], other isolates from these same farms were classified as part of clade A, and VCG 0120 isolate Foc 229A was characterized among the group of isolates belonging to VCG 0120, which is usually associated with Foc ST4, essentially eliminating the possibility of this being an isolate of TR4. In addition, its variation in terms of virulence and aggressiveness noted here may represent the existence of genetic diversity, even among individuals of the same race [69,70,71].

The better performance of the BRS Platina cultivar can be explained by its genealogy, considering that it is a tetraploid hybrid (AAAB) developed from the cross between the Prata-Anã cultivar (AAB) and the improved diploid M53 (AA) [72,73]. Diploid M53 was notable for not showing symptoms of Fusarium wilt in the field and for being the parent of other hybrids, such as BRS Princesa, BRS Preciosa and BRS Pacovan Ken, which are already widespread in the domestic market of Brazil because of their good agronomic and sensory characteristics [74,75,76,77]. The M53 hybrid was also characterized as resistant to tropical Foc race 4 in a resistance test that was performed in Northern Territory (Australia) in an area naturally infested with the pathogen, with no symptoms of Fusarium wilt being observed during the culture under high inoculum pressure [78]. This information is important because it allows assessments of a possible resistance of the BRS Platina cultivar to TR4. In partnership with the Colombian Agricultural Research Corporation (AgroSavia, Bogotá – Mosquera, Colombia), the BRS Platina cultivar is being quarantined in that country and subsequently will be challenged in the presence of Foc TR4 to confirm its putative resistance.

Although Foc TR4 has not yet been detected in Brazil, its recent introduction in Colombia and Peru further demonstrates the need for prevention of a possible introduction [11,79]. In addition, there is much damage caused by isolates of the Foc population in Brazil even in the absence of TR4, where it was shown that the average incidence of Fusarium wilt is 11%, causing an estimated productivity loss of 1.8 t ha-1 year-1 [80]. Thus, the data presented here suggest that the BRS Platina cultivar has potential for use in breeding programs focused on resistance to Foc TR4.

## 4. Materials and Methods

### 4.1. Foc Isolates

Three Foc isolates from the biological collection of the Laboratory of Plant Pathology of Empresa Brasileira de Pesquisa Agropecuária (EMBRAPA) were selected for this study. Isolate 0801 is characterized as the standard for race 1 (R1), and isolate 218A is part of the vegetative compatibility group VCG 0120, characterized as subtropical race 4 (ST4) [66]. The 229A isolate was selected due to the percentage of Fusarium wilt symptoms caused in the Grand Naine cultivar in a previous study. In the mentioned study, this isolate was considered virulent, defined as the ability to cause disease in a given cultivar [67].

Each isolate was grown in potato dextrose agar culture medium and subsequently multiplied in rice (20 mL of spore suspension in 500 g of autoclaved rice). After growth in this culture medium, the inoculum went through a serial dilution and an aliquot of each dilution was deposited in Petri dishes. After 48 h, the cfu (colony-forming units) were counted in a colony counter under a stereomicroscope and the inoculum adjusted to the concentration of 10^6^ conidia/g of rice. 

### 4.2. Plant Material

Three banana cultivars with different responses to Foc were used: the cultivar Prata-Anã (subgroup Prata; AAB), which is susceptible to Foc R1, ST4 and 229A; the cultivar Grand Naine (subgroup; Cavendish; AAA), which is resistant to R1, moderately resistant to ST4 and possibly susceptible to Foc 229A; and the cultivar BRS Platina (subgroup Prata; AAAB), which is resistant to R1 and whose response to ST4 and 229A has not yet been evaluated. 

Plantlets of each cultivar grown from tissue culture were acclimated under ideal growth conditions and planted in pots with substrate consisting of pine bark (Tecnomax^®^) and coconut fiber (5:1; *v*:*v*), where they remained for 40 days under adequate irrigation and fertilization. Fertilization was carried out according to banana plant recommendations with application of limestone, monoammonium phosphate (MAP), potassium chloride, ammonium sulfate and FTE BR 12, a compost formulated with micronutrients. The temperatures varied between 25 and 30 °C, with relative humidity of approximately 60 to 80% and 12:12 h photoperiod.

### 4.3. Bioassay

For inoculation in the greenhouse, 40-day-old plants were removed from the pots, and the substrate was carefully removed from the roots. Subsequently, the substrate was infested with 40 g of the inoculum of each Foc isolate separately, and subsequently, the plants were replanted in the pots, placing the roots in contact with the infested substrate. The experiment was conducted in a completely randomized design, with nine treatments: 3 cultivars x 3 Foc isolates.

For each cultivar, 40 plants were inoculated, and only autoclaved rice was deposited in the pots holding control plants. Three root-collection times were established for the subsequent analyses (12, 24 and 48 h after inoculation (HAI)), and a collection was performed at 30 days after inoculation. For each treatment, three biological replicates were collected at each collection time. At the end of the experimental period, 90 days after inoculation, 10 plants from each treatment remained for evaluation of internal symptoms of Fusarium wilt, using the methods described by Dita et al. [28]. The scores obtained in the evaluation of symptoms were converted into an internal index according to the McKinney [81] formula.

### 4.4. RNA Extraction and cDNA Synthesis

Root samples used for total RNA extraction were collected, immediately immersed in liquid nitrogen and stored in an ultrafreezer (−80 °C) until processing. Plants collected at 12, 24 and 48 HAI and control plants were used for total RNA extraction using a cetyltrimethylammonium bromide (CTAB 2%) protocol previously described by Zhao et al. [82]. RNA was quantified by comparative analysis on a 1% agarose gel and by spectrophotometry with a Nanodrop ND-2000 device (Thermo Scientific, Waltham, MA, USA). The RNA samples were treated with DNase (RNase TURBOfree-Ambion), and cDNA synthesis was performed with a high-capacity RNA-to-cDNA Kit (Applied Biosystems, Waltham, Massachusetts, EUA) following the manufacturer’s recommendations.

### 4.5. Gene Expression Analysis by Quantitative Real-Time PCR

The genes listed in Table 1, Cellulose synthase A catalytic subunit 7 (CESA7), Auxin transporter-like protein 1 (ATL) and Putative Disease resistance response protein 206 (PI206), were derived from a previous study that analyzed the transcriptomic profiles of the cultivars BRS Platina, Prata-Anã and Silk via RNAseq in response to infection by Foc R1. After large-scale analysis of the data obtained by RNAseq, these genes were selected because they are related to plant defense responses and shared among cultivars or exclusive to some cultivars [32]. The gene encoding transcription factor WRKY 22 (WRKY22), pathogenesis-related protein 1 (PR1), chitinase (CHI) and lipoxygenases (LOX) were derived from a study that compared infection processes and gene expression levels in a cultivar of banana (Cavendish) inoculated with Foc 1 and Foc TR4 [27]. 

The real-time PCR assays were performed in the Applied Biosystems 7300 Real-Time PCR System (ABI, Foster City, CA, USA) using SYBR Green PCR mix (Ludwig Biotech, Alvorada-RS, Brazil) with the primers listed in Table 1. The reaction mixture included 1 µL of cDNA, 0.3 μL of each primer (RF), 5 μL of SYBR Green PCR mix and 3.4 μL of nuclease-free water for a total volume of 10 µL in each reaction. The amplification conditions of the reactions were as follows: 50 °C for 2 min and 95 °C for 10 min followed by 40 cycles of denaturation at 95 °C for 15 s and annealing and primer extension at 58 °C for 1 min. The 25S gene was used as an endogenous reference gene, as previously tested [46,63]. Each biological replicate was examined in triplicate using a relative quantification analysis and the pairwise fixed reallocation randomization test method, where the values of the quantification cycle (Ct) were used to calculate the relative quantities by the 2 (ΔΔ TC) formula [84,85]. Therefore, the values of the Qc’s (quantification cycles) used to determine the gene differential expression between the inoculated treatments and the control for the different time periods were determined by the expression based on the PCR exponential reaction, RQ = 2 -ΔΔCT, whereas RQ (relative quantification) determines the level of gene expression; the CT indicates the amplification cycle; ΔCT expresses the difference between the CT of the amplified sample for the specific gene and the CT of the same sample amplified for the normalizing gene (reference gene). Hence, the ΔΔCt represents the difference between the ΔCT of the sample of interest at a certain time period and the ΔCT of the control sample (non-inoculated plants) [84]. All the values are expressed as the mean ± standard deviation.

### 4.6. Histological and Histochemical Analyses

Small root fragments were collected and immediately immersed in Karnovsky’s solution [86], where they remained for 48 h; the fragments were then dehydrated in an increasing ethanol series with an interval of three hours each (30–100%). Infiltration and blocking were performed with a Historesin kit (hydroxyethyl methacrylate, Leica Heldelberg, Germany). After Historesin polymerization, histological sections (8 μm) were obtained with a Leitz 1516 microtome. The sections were mounted on histological slides that were stained with ferric chloride for three hours to detect phenolic compounds [87] and Calcofluor White dye 0.01% to detect cellulose [88]. The histological sections were analyzed and photographed under a B x S1 fluorescence microscope (Olympus Latin America Inc., Tokyo, Japan).

Analysis of root clarification and staining of fungal structures were performed according to a method described Phillips and Haymann [89] with modifications. Briefly, for clarification, the roots were immersed in a 10% KOH (potassium hydroxide) solution at room temperature for 48 h and then in a 1% HCl solution for 30 min, and the dye was used to stain the structures. Trypan blue in 0.05% solution (2:1:1 lactic acid: glycerol: water) was applied for 1 h. After staining, slides were prepared, and the fragments microphotographed under a light microscope (Olympus Latin America Inc., Tokyo, Japan).

### 4.7. Analysis by Scanning Electron Microscopy—SEM

After dehydration in an ethanol series, root samples were dried in a critical point apparatus (Leica EM CPD 030) using liquid CO_2_. Samples were fixed to a metallic support (stubs) with double-sided carbon adhesive tape and metalized with gold in JEOL Smart Coater equipment (DII-29010SCTR). The observations and electron micrographs were performed in a JEOL JSM-6390LV scanning electron microscope in the electron microscopy laboratory of the Gonçalo Moniz Institute, Fiocruz, Salvador-BA. 

## 5. Conclusions

In this study, the histological, histochemical and molecular analysis of the roots of the banana cultivars BRS Platina, Grand Naine and Prata-Anã and the differences in the defense response profiles between cultivars infected by these three Foc isolates with different virulence patterns were compared. Results showed that the CESA7, ATL, PI206, WRKY22, PR1, CHI and LOX genes, as well as the increased presence of cellulose, phenolic compounds and calcium oxalate crystals, were induced in BRS Platina, suggesting their important roles in incompatible interactions between resistant banana cultivars and Foc ST4 and R1. However, these defense responses were suppressed or reduced mainly by an isolate with greater virulence in Prata-Anã and Grand Naine, suggesting that these strategies are not adopted by these cultivars and that this isolate can suppress these defense responses as part of their infection strategies. Additional studies are needed to determine the functions of the genes studied and the corresponding pathways. In conclusion, our study expands the information on compatible and incompatible interactions between banana genotypes and the Foc pathogen and highlights the understanding of the response mechanism of the BRS Platina cultivar to Foc TR4 as the next phase of our research.

## Figures and Tables

**Figure 1 plants-11-02339-f001:**
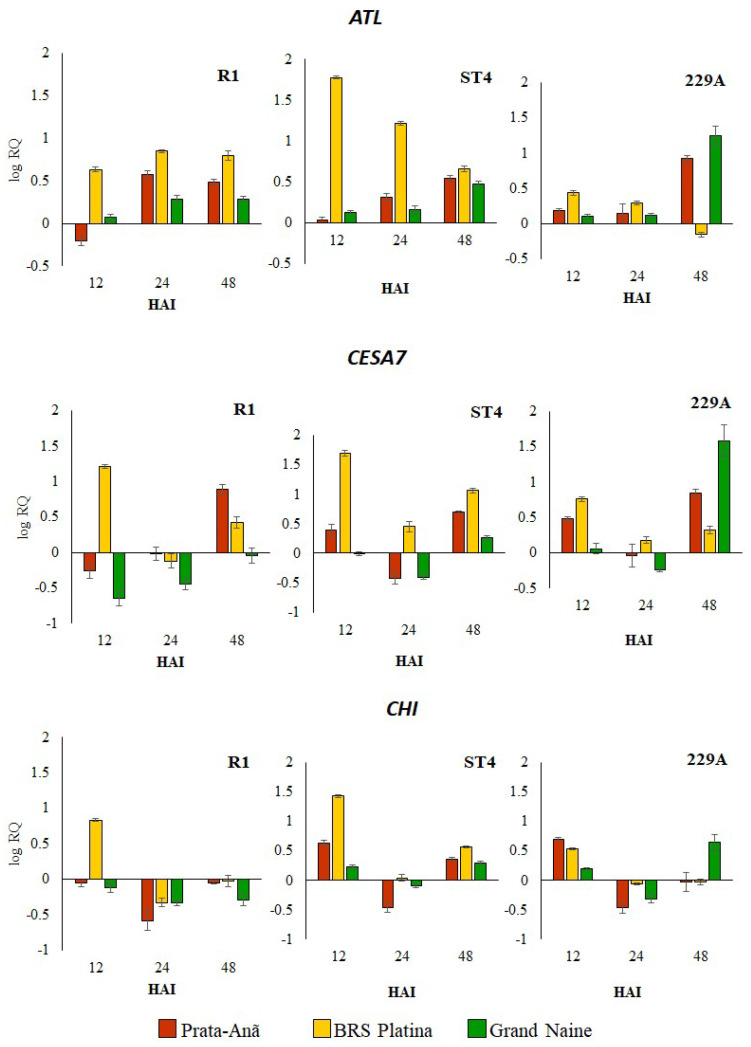
Relative expression levels of three defense-related genes in banana genotypes at 12, 24 and 48 h after inoculation (HAI) with *Fusarium oxysporum* f. sp. *cubense* races 1 (Foc R1), subtropical 4 (Foc ST4) and isolate 229A. Levels of the relative expression of the defense-related genes Auxin transporter-like protein 1 (ALT), cellulose synthase A catalytic subunit 7 (CESA7) and chitinase (CHI) were analyzed by RT-qPCR. The data represent the means ± standard deviations of three biological replicates and three technical replicates. QR values correspond to genes with differential expression at 12, 24 and 48 h after inoculation (HAI) compared to 0 HAI (non-inoculated plants).

**Figure 2 plants-11-02339-f002:**
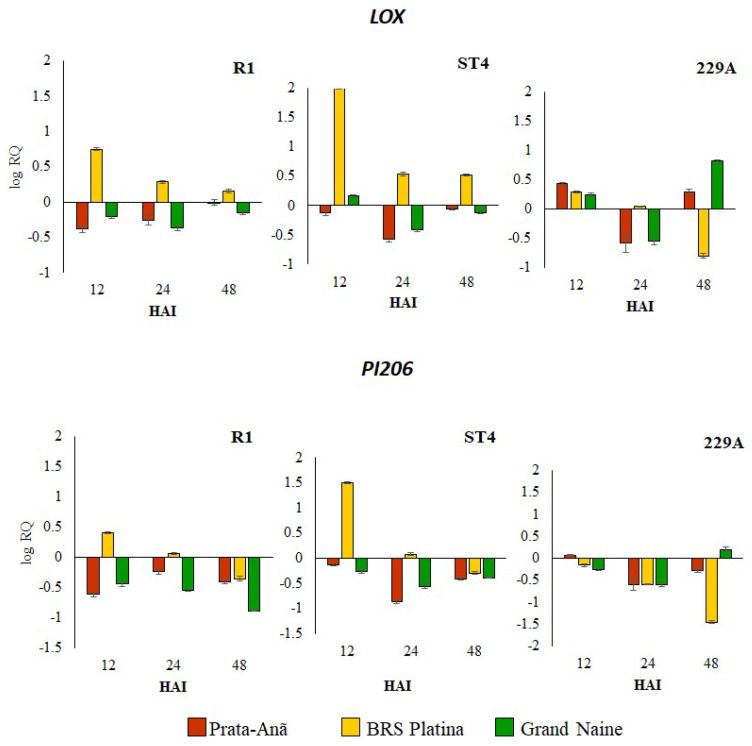
Relative expression levels of two defense-related genes in banana genotypes at 12, 24 and 48 h after inoculation (HAI) with *Fusarium oxysporum* f. sp. *cubense* races 1 (Foc R1), subtropical 4 (Foc ST4) and isolate 229A. Levels of relative expression of the genes related to the defense lipoxygenase (LOX) and putative disease resistance response protein 206 (PI206) were analyzed by RT-qPCR. The data represent the means ± standard deviations of three biological replicates and three technical replicates. QR values correspond to genes with differential expression at 12, 24 and 48 h after inoculation (HAI) compared to 0 HAI (non-inoculated plants).

**Figure 3 plants-11-02339-f003:**
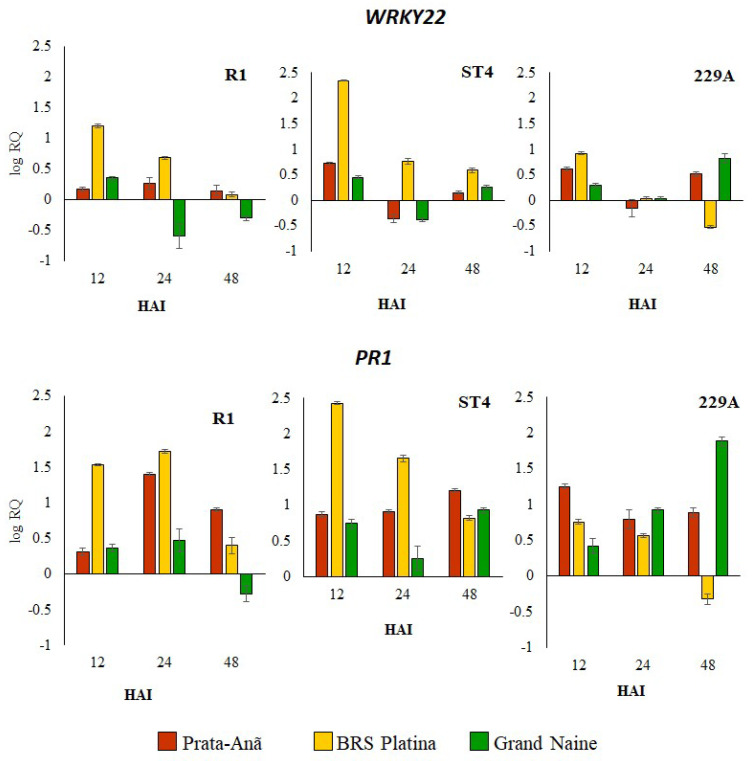
Relative expression levels of two defense-related genes in banana genotypes at 12, 24 and 48 h after inoculation (HAI with *Fusarium oxysporum* f. sp. *cubense* races 1 (Foc R1), subtropical 4 (Foc ST4) and isolate 229A. The relative expression levels of the defense-related gene transcription factor WRKY22 and protein-related pathogenesis 1-like PR1 were analyzed by RT-qPCR. The data represent the means ± standard deviations of three biological replicates and three technical replicates. QR values correspond to genes with differential expression at 12, 24 and 48 h after inoculation (HAI) compared to 0 HAI (non-inoculated plants).

**Figure 4 plants-11-02339-f004:**
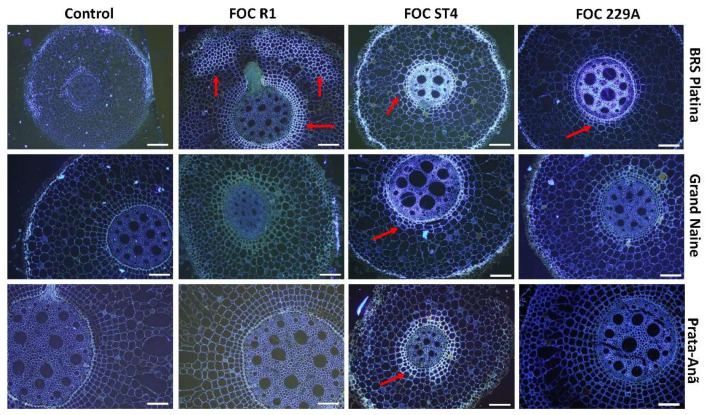
Cross-sectional fluorescence micrographs of roots of banana cultivars with xylem cavities infested by different *Fusarium oxysporum* f. sp. *cubense* strains 12 h after inoculation (HAI) and stained with Calcofluor White to detect cellulose. Red arrows indicate second-grade white-blue fluorescence related to the concentration of cellulose in the tissues. Foc: *Fusarium oxysporum* f. sp. *Cubense*; R1: race 1; ST4: subtropical race 4; Bars = 200 μm.

**Figure 5 plants-11-02339-f005:**
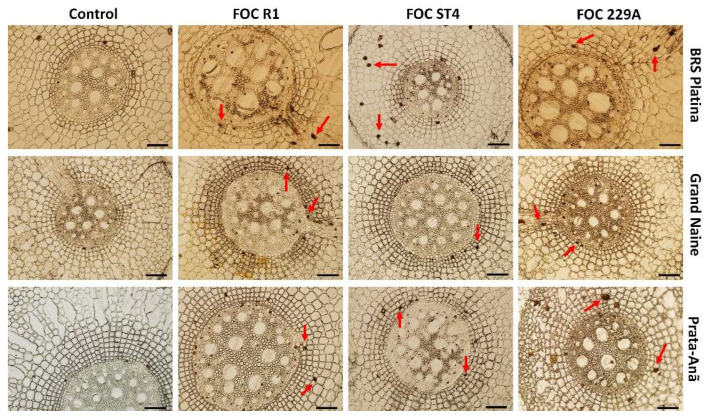
Cross-sectional micrographs of roots of banana cultivars with xylem cavities infested by different *Fusarium oxysporum* f. sp. *cubense* strains 12 h after inoculation (HAI) and stained with ferric chloride for the detection of phenolic compounds. Red arrows indicate sections with accumulation of phenolic compounds. Foc: *Fusarium oxysporum* f. sp. *cubense.* R1: race 1. ST4: subtropical race 4. bars = 200 μm.

**Figure 6 plants-11-02339-f006:**
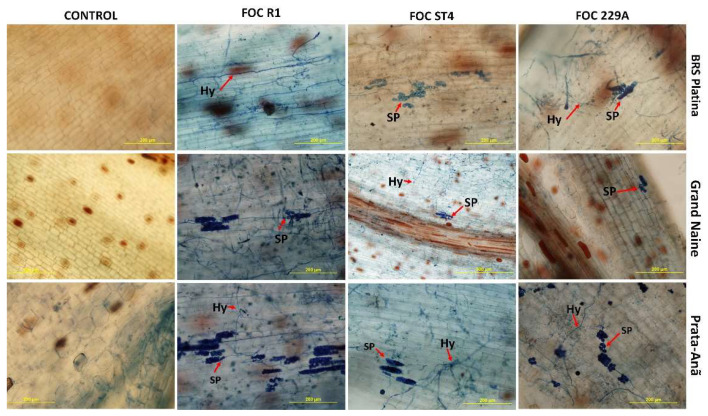
Root clarification and staining of fungal structures with trypan blue dye after infestation by different isolates of *Fusarium oxysporum* f. sp. *cubense* 90 days after inoculation. SP: spores; HY: hyphae. Red arrows indicate hyphae (Hy) and spores (SP). Foc: *Fusarium oxysporum* f. sp. *cubense*; R1: race 1; ST4: subtropical race 4; bars = 200 μm.

**Figure 7 plants-11-02339-f007:**
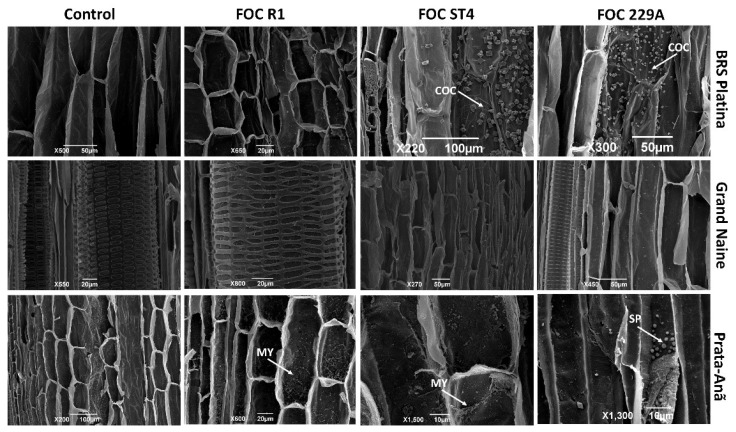
Scanning electron micrographs of longitudinal sections of roots of banana cultivars with xylem cavities infested by different *Fusarium oxysporum* f. sp. *cubense* isolates 48 h after inoculation (HAI). Foc: *Fusarium oxysporum* f. sp. *cubense*; R1: race 1; ST4: subtropical race 4; COC: calcium oxalate crystals; MY: mycelium; SP: spores.

**Figure 8 plants-11-02339-f008:**
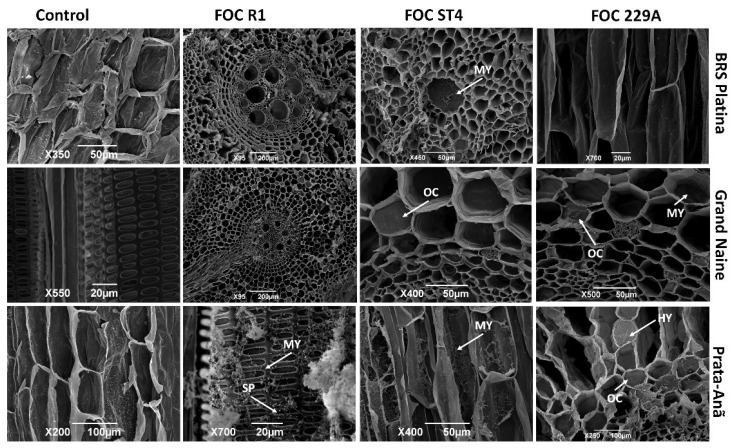
Scanning electron micrographs of cross-sectional and longitudinal sections of roots of banana cultivars with xylem cavities infested by different *Fusarium oxysporum* f. sp. *cubense* strains at 30 days after inoculation (DAI). Foc: *Fusarium oxysporum* f. sp. *cubense*; R1: race 1; ST4: subtropical race 4; MY: mycelium; SP: spores; HY: hyphae.

**Figure 9 plants-11-02339-f009:**
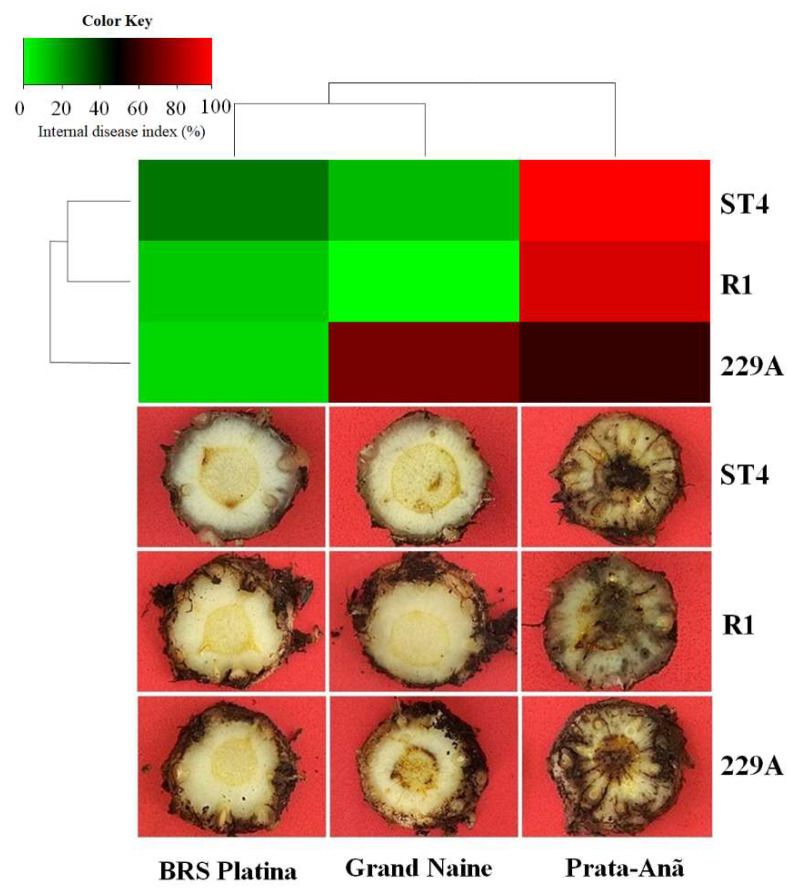
Heat map of internal symptom indices and cross-sections of the rhizomes of three banana cultivars 90 days after inoculation (DAI) with *Fusarium oxysporum* f. sp. *cubense* isolates that differ in virulence. The heat map colors reflect high levels of internal symptoms (intense red) and low levels of internal symptoms (intense green). The disease indices were calculated from the evaluation scores of 10 plants of each cultivar using the Dita et al. 2014 [28] rating scale. Foc: *Fusarium oxysporum* f. sp. *cubense*; R1: race 1; ST4: subtropical race 4.

**Table 1 plants-11-02339-t001:** Primers used in gene expression analysis of the interaction of banana cultivars and *Fusarium oxysporum* f. sp. *cubense* with different levels of virulence.

ID	Gene	Description	Sequence (5′-3′)	pb	Reference
GSMUA_Achr5T15720_001	CESA7	Cellulose synthase A catalytic subunit 7	F: GAGAATGGAGAACGGGTGCA	108	[32]
			R: CCCCTCCATGTCTCTCTCCA		
GSMUA_Achr8T02300_001	ATL	Auxin transporter-like protein 1	F: GGTTCAGCTGCTCCTCCAAT	161	[32]
			R: AGAACAGCTGCAGGATCACC		
GSMUA_Achr8T15700_001	PI206	Putative Disease resistance response protein 206	F: AGTACAACGGGAGCAGCTTC	128	[32]
			R: GATGAGCCTGCTGATGGTGT		
XM_009417035.2	WRKY22	Transcription factor WRKY 22	F: CGTGACGTACGAAGGAGAGCA	95	[27]
			R: GGTCAACGCGAAGTCAACCA		
XM_009417035.2	PR1	Pathogenesis-related proteins 1	F: AGTTATGGACGAGCTACCCG	77	[27]
			R: GTAGCTGAAGTACTTCCCCTC		
XM_009415745.2	CHI	Chitinase	F: TACTGGAACTACAACTACGGAGC	82	[27]
			R: CGTTCTGCTCGAGGTACTC		
XM_008803483.2	LOX	Lipoxygenases	F: ACGATGCAGACGGTATTGGAGT	94	[27]
			R: GGTACTGTCCGAAGTTGACG		
25SMU	25S	25S rRNA	F: ACATTGTCAGGTGGGGAGTT	106	[83]
			R: CCTTTTGTTCCACACGAGATT		

## Data Availability

All data were collected at Embrapa Mandioca e Fruticultura, Brazil.

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
