# Peer review of "Molecular, Histological and Histochemical Responses of Banana Cultivars Challenged with Fusarium oxysporum f. sp. cubense with Different Levels of Virulence"

_plants, 2022, doi:10.3390/plants11182339_

Round 1

Reviewer 1 Report

The paper "Molecular, histological and histochemical responses of banana cultivars challenged with Fusarium oxysporum f. sp. cubense with different levels of virulence" by Rocha et al. is interesting and thorough. I do not see any major issues with the paper. Overall, the research is sound, and the figures are well presented. I think the English could be improved slightly in the introduction. Line changes are listed below.

Line changes

38 this opening sentence not quite relevant to the paper. Thousands of families?, is there a citation for this?

54 unnecessary wording, are the citations referring to global warming or to the effect of the stress on the plant?

66-74 not really needed, this is not relevant to the main topic just a list of centers

160 of three replicate .. experiments or bananas?, please clarify

Fig.4, this is a very nice figure overall, however, it is hard to see differences among treatments. Could you note where the damage/occlusions from fusarium are in the roots?

Fig.7. which spores are they? Microconidia? Hard to tell by figure alone.

Fig.9. this is a great heat map, very clearly shows results.

396 is this specific to fusarium or all pathogens? What about occlusions and phytoalexins?

603 how was the number of conidia per rice grain measured?

611 what type of light and fertilizer were used? How big were the plants when used?

683 brand name? 

Author Response

Comments and Suggestions for Authors

The paper "Molecular, histological and histochemical responses of banana cultivars challenged with Fusarium oxysporum f. sp. cubense with different levels of virulence" by Rocha et al. is interesting and thorough. I do not see any major issues with the paper. Overall, the research is sound, and the figures are well presented. I think the English could be improved slightly in the introduction. Line changes are listed below.

Line changes

38 this opening sentence not quite relevant to the paper. Thousands of families?, is there a citation for this?

A = The sentence was removed from the text.

54 unnecessary wording, are the citations referring to global warming or to the effect of the stress on the plant?

A = The sentence was removed from the text. The citations refer to the effect of stress in the plants.

66-74 not really needed, this is not relevant to the main topic just a list of centers.

A = We agree with the reviewer´s comment and the paragraph was removed from the text.

160 of three replicate.. experiments or bananas? please clarify.

A = The data comprises three biological banana replicates. An explanatory sentence was added to the text for better understaning.

Fig.4, this is a very nice figure overall, however, it is hard to see differences among treatments. Could you note where the damage/occlusions from fusarium are in the roots?

A = Figure 4 depicts a cross-section of roots stained with Calcofluor White which detects cellulose. The red arrows show the presence of a white-blueish color which indicates the presence of this compound under fluorescent microscopy, but it is not possible to see occlusion nor any damage to Fusarium with this evaluation, only by Scan electron microscopy (SEM) in figures 7 and 8.  

Fig.7. which spores are they? Microconidia? Hard to tell by figure alone.

A= We opted for not classifying the type of spore found. Since it was not possible to make a more detailed characterization, the result was only compared to control plants of each cultivar which did not show these structures of the pathogen.

Fig.9. this is a great heat map, very clearly shows results.

396 is this specific to fusarium or all pathogens? What about occlusions and phytoalexins?

A = This can be seen for other pathogen x host interactions, not being specific for Fusarium.

603 how was the number of conidia per rice grain measured?

A = Information added to the text accordingly.

611 what type of light and fertilizer were used? How big were the plants when used?

A = Information added to the text accordingly.

683 brand name?

A = Information added to the text accordingly.

Reviewer 2 Report

The one thing I think must be learned up  is an explanation of how the relative values of the RT-PCR averages were determined.  Were they compared to samples collected from the mock inoculation plants?  If not, what was the basis for up and down regulation? 

Author Response

Comments and Suggestions for Authors

The one thing I think must be learned up is an explanation of how the relative values of the RT-PCR averages were determined. Were they compared to samples collected from the mock inoculation plants? If not, what was the basis for up and down regulation?

A = We used the data from each control to calculate the RQ values at each time separately and thus obtained the base value for ascending and descending expression of the treatments. We have inserted a more detailed explanation in the Material and Methods section.